

# Soil mineralized carbon drives more carbon stock in coniferous-broadleaf mixed plantations compared to pure plantations

Zhenzhen Hao[1,2,3], Zhanjun Quan[3], Yu Han[3], Chen Lv[4], Xiang Zhao[5], Wenjie Jing[3], Linghui Zhu[6] and Junyong Ma[1,7]

[1] Key Laboratory of Mine Ecological Effects and Systematic Restoration, Ministry of Natural Resources, Beijing, China
[2] State Key Laboratory of Environmental Criteria and Risk Assessment, Chinese Chinese Research Academy of Environmental Sciences, Beijing, China
[3] State Key Laboratory of Grassland Agro-Ecosystem, Institute of Arid AgroEcology, School of Life Sciences, Lanzhou University, Lanzhou, China
[4] Xichuan County Water Conservancy Bureau, Henan, China
[5] State Environmental Protection Key Laboratory of Estuarine and Coastal Environment, Water Research Institute, Chinese Research Academy of Environmental Sciences, Beijing, China
[6] School of Soil and Water Conservation, Beijing Forestry University, Beijing, China
[7] Key Laboratory of Ministry of Forest Cultivation and Conservation of Ministry of Education, Beijing Forestry University, Beijing, China

Corresponding author
Junyong Ma,
mjy172404707@me.com

## ABSTRACT

Forest soil carbon (C) sequestration has an important effect on global C dynamics and is regulated by various environmental factors. Mixed and pure plantations are common afforestation choices in north China, but how forest type and environmental factors interact to affect soil C stock remains unclear. We hypothesize that forest type changes soil physicochemical properties and surface biological factors, and further contributes to soil active C components, which together affect soil C sequestration capacity and C dynamic processes. Three 46-year-old 25 m × 25 m pure *Pinus tabulaeformis* forests (PF) and three 47-year-old 25 m × 25 m mixed coniferous-broadleaf (*Pinus tabulaeformis-Quercus liaotungensis*) forests (MF) were selected as the two treatments and sampled in August 2016. In 2017, soil temperature (ST) at 10 cm were measured every 30 min for the entire vegetation season. Across 0–50 cm (five soil layers, 10 cm per layer), we also measured C components and environmental factors which may affect soil C sequestration, including soil organic carbon (SOC), soil total nitrogen (STN), dissolved organic carbon (DOC), microbial biomass carbon (MBC), soil moisture (SM) and soil pH. We then incubated samples for 56 days at 25 °C to monitor the C loss through $CO_2$ release, characterized as cumulative mineralization carbon (CMC) and mineralized carbon (MC). Our results indicate that ST, pH, SM and litter thickness were affected by forest type. Average SOC stock in MF was 20% higher than in PF (MF: 11.29 kg m$^{-2}$; PF: 13.52 kg m$^{-2}$). Higher CMC under PF caused more soil C lost, and CMC increased 14.5% in PF (4.67 g kg$^{-1}$ soil) compared to MF (4.04 g kg$^{-1}$ soil) plots over the two-month incubation period. SOC stock was significantly positively correlated with SM ($p < 0.001$, $R^2 = 0.43$), DOC ($p < 0.001$, $R^2 = 0.47$) and CMC ($p < 0.001$, $R^2 = 0.33$), and significantly negatively correlated with pH

($p < 0.001$, $R^2 = -0.37$) and MC ($p < 0.001$, $R^2 = -0.32$). SOC stock and litter thickness may have contributed to more DOC leaching in MF, which may also provide more C source for microbial decomposition. Conversely, lower SM and pH in MF may inhibit microbial activity, which ultimately makes higher MC and lower CMC under MF and promotes C accumulation. Soil mineralized C drives more C stock in coniferous-broadleaf mixed plantations compared to pure plantations, and CMC and MC should be considered when soil C balance is assessed.

# INTRODUCTION

The turnover of soil carbon (C) and C processes have been changed by forest ecosystem structure and functional dynamic, which may play an important role in global climate change (*Wang et al., 2022*; *Ma et al., 2022*). Half of the global forest C is stored in soils (*Pan et al., 2011*) with soil organic carbon (SOC) stock estimated to be over three times the size of atmospheric stock and four times that of biotic stock (*Lal, 2004*). SOC stock in forests may influence atmospheric $CO_2$ concentrations and regulate the greenhouse effect (*Amundson, 2001*). In addition, the large soil reservoir is not permanent, but result from a dynamic equilibrium between organic and inorganic matter entering and leaving the soil (*Soucémarianadin et al., 2018*; *Tian et al., 2016*). Accurate determination of changes in SOC stocks and response analysis of dominant environmental factors are prerequisite to understand the role of soils in the global C cycling and to verify changes in stocks due to management.

Afforestation has been the most influential human activity in altering forest ecosystem structure and function that has been implemented worldwide (*IPCC, 2022*). Forest management has received increasing attention because of its predictable effects on ecosystems, specifically through C sequestration (*Fang et al., 2001*; *Richter et al., 1999*). Research indicates that the C sequestration capacity of soil is influenced by forest management, which differs depending on tree species composition (*Galka et al., 2014*), forest stand age (*Lucas-Borja et al., 2016*), forest density (*Ma et al., 2018*; *Segura et al., 2017*) and other forest variables (*Verkerk, de Arano & Palahí, 2018*; *Li et al., 2018*; *Chen & Shrestha, 2012*). The afforestation method of transitioning from pure forest to mixed forest is widely used and has garnered increasing attention (*Bravooviedo et al., 2014*; *Pretzsch, Schütze & Uhl, 2013*). Although soil C sequestration capacity between mixed and pure forests has been studied (*Cremer, Kern & Prietzel, 2016*; *Wang et al., 2014*), the mechanisms to explain the differences still need to be further explored because many environmental factors are involved in these dynamic processes.

The soil C dynamics of forests may be impacted by interacting environmental, and physical factors such as soil temperature (ST) (*Uvarov, Tiunov & Scheu, 2006*), soil moisture (SM) (*Yoon et al., 2014*), soil texture (*Cai et al., 2016*) and bulk density (*Vos et al., 2005*); chemical factors such as soil pH (*Motavalli et al., 1995*) and elemental nitrogen and

phosphorus (*Liu et al., 2015*); and bioenvironmental factors such as plant diversity (*Chen et al., 2018*), tree age (*Ma et al., 2018*), litter (*Tan & Chang, 2007*) and root matter (*Hertel & Leuschner, 2002*). The species composition of the aboveground vegetation will affect the quantity and quality of surface litter and root exudates and the input processes of organic C source (*Chen & Xu, 2008*; *Tan & Chang, 2007*). ST, SM and pH can affect the export process of soil C, such as soil respiration, soil C transfer and loss (*Uvarov, Tiunov & Scheu, 2006*; *Yoon et al., 2014*). Simultaneously, *Schrumpf et al. (2011)* research shown that soil SOC stocks were calculated based on SOC concentrations, bulk densities, and the fine earth fractions, and in undisturbed forest soils with low stone contents, SOC concentrations contributed most to SOC stock variability. We should fully consider the climatic, land-use, and soil types of the study site, and then comprehensively select the formula for calculating soil SOC stock to further analyze the response mechanisms of dynamic process of C accumulation and loss to multiple environmental factors.

Soil active C refers to the part of SOC with poor stability, quick turnover, easy mineralization and decomposition, and high activity to plants and soil microorganisms, among which dissolved organic carbon (DOC), microbial biomass carbon (MBC) and mineralized carbon (MC) are important indicators (*Tian et al., 2015*; *Wang et al., 2014*; *Hu et al., 1997*). Studies have shown that active characteristics make soil-activated C vulnerable to environmental factors (*Zhang et al., 2015*; *Jiang et al., 2006*), and active C can reflect small changes in SOC caused by management measures or climate change (*Leifeld & Kögel-Knabner, 2005*), which plays an important role in soil C sequestration capacity and greenhouse gas emissions (*Liang et al., 2012*; *Yang et al., 2009*). DOC is an organic C source that can be directly used by soil microorganisms and is active in the physical movement and chemical transformation of soil (*Chen et al., 2018*). Meanwhile, DOC leaching is also an important mechanism of SOC loss (*Oliveira et al., 2016*). MBC is the most active component of SOC, revealing microbial activity and concentration in soil and is an important indicator for measuring soil fertility (*Xu, Inubushi & Sakamoto, 2006*). Above-ground vegetation type is generally considered to be an important factor affecting microbial activity (*Pötzelsberger & Hasenauer, 2015*). CMC is the amount of $CO_2$ released after SOC was mineralized into inorganic C in a certain of time (measured by g kg$^{-1}$ release of $CO_2$-C), and MC was the proportion of $CO_2$-C content released by SOC mineralization to soil total organic C content in a certain period of time (%) (*Sanford & Kucharik, 2013*). The amount (CMC) and intensity (MC) of $CO_2$ released from SOC mineralization by microbial decomposition can reflect the amount, activity and species of microorganisms and can be used to evaluate the influence of environmental factors or human factors on soil (*Paul, Morris & Bohm, 2001*). When studying C sequestration in forest ecosystems, individual C stocks in soil can provide insight into the mechanisms favoring soil C turnover and persistence.

The objectives of this study are three-fold: (i) to determine C stock and active components between PF and MF; (ii) to evaluate which forest type can maintain a better C sequestration strategy and (iii) to reveal the potential mechanism of C dynamics between the two forest types through variation in active C components and environmental factors.
**Table 1 Basic characteristics of the plots.**

| Treatment | Plot | Dominant tree species | Age years | Elevation m | Slope ° | Aspect | Tree height m | DBH cm |
|---|---|---|---|---|---|---|---|---|
| PF | NO. 1 | *Pinus tabuliformis* | | 46 | 1,906 ± 21 | 24.5° | Nothwest | 14.91 ± 3.80 | 19.53 ± 7.90 |
| | NO. 2 | *Pinus tabuliformis* | | 46 | 1,869 ± 21 | 23.5° | Nothwest | 16.21 ± 4.00 | 19.21 ± 6.60 |
| | NO. 3 | *Pinus tabuliformis* | | 46 | 1,853 ± 9 | 22° | Nothwest | 16.85 ± 3.90 | 19.70 ± 6.00 |
| MF | NO. 4 | *Pinus tabuliformis* | *Quercus wutaishansea Mary* | 47 | 1,628 ± 8 | 28.8° | Nothwest | 14.31 ± 5.50 | 16.12 ± 7.90 |
| | NO. 5 | *Pinus tabuliformis* | *Quercus wutaishansea Mary* | 47 | 1,607 ± 20 | 28.5° | Nothwest | 15.46 ± 10.10 | 13.76 ± 6.10 |
| | NO. 6 | *Pinus tabuliformis* | *Quercus wutaishansea Mary* | 47 | 1,647 ± 11 | 25.5° | Nothwest | 15.77 ± 5.40 | 16.37 ± 4.70 |

| Living branch height m | BD g cm$^{-3}$ | Mechanical composition (%) | | |
|---|---|---|---|---|
| | | <0.002 mm | 0.002–0.05 mm | 0.05–2.00 mm |
| 7.81 ± 3.20 | 1.17 ± 0.11 | 22.83 | 37.04 | 40.13 |
| 8.70 ± 2.20 | 1.24 ± 0.09 | 19.13 | 32.37 | 48.5 |
| 9.39 ± 2.40 | 1.37 ± 0.03 | 25.23 | 28.19 | 46.58 |
| 6.27 ± 4.10 | 1.27 ± 0.11 | 16.26 | 29.34 | 54.4 |
| 6.14 ± 3.90 | 1.26 ± 0.10 | 21.45 | 33.93 | 44.62 |
| 10.15 ± 3.30 | 1.34 ± 0.11 | 18.32 | 32.64 | 49.04 |

**Note:**
Soil characteristics of the studied stands represent the average values for soil depth of 0–50 cm, with standard error. DBH, diameter at 1.2 m breast height; BD, bulk density. All the basic information was measured in August of 2016 (means ± SD, $n = 3$).

We hypothesized that MF soil have better C sequestration capacity and lower $CO_2$ released from the soil of MF derived by some active C components.

## MATERIALS AND METHODS

### Study site description and experimental design

We performed our study in August 2016 at the Taiyue Mountain Ecosystem Research Station (CFERN) in a continental seasonal climate zone of Shanxi province, Taiyue Mountain, North China (112°01′–112°15′E, 36°31′–36°43′N; elevation 1,607–1,906 m above sea level). The mean monthly temperature of this region is highest in July (17.4 °C) and lowest in January (−10.4 °C). Precipitation mostly falls from July to September, and the mean annual precipitation ranges between 600 and 650 mm (*Ma, Han & Cheng, 2020*).

Two types of soil and their respective C stocks were measured. One treatment was the pure forest (PF—*Pinus tabulaeformis* forest only), which was planted in 1970 and has remained unchanged since. The other forest treatment was the mixed forest (MF—*Pinus tabulaeformis-Quercus liaotungensis* mixed forest), which was planted in 1969. We established three 25 m × 25 m plots in each of the PF and MF forests in August 2016. We left 5-m gaps among three repeated plots, and plots of same treatment were located in similar elevations, slopes, and aspects (Table 1).

Above-ground forest information, including dominant tree species, tree height, DBH (diameter at 1.2 m breast height), living branch height, were also recorded and are presented in Table 1. Basic soil characteristics were measured in August 2016, including soil pH, bulk density (BD), SM, SOC, STN, DOC, DON, MBC, MBN, and mechanical composition (clay, silt, sand).

**Table 2 Plant species diversity in arborous, shrub and herbaceous layer of the plots and litter thickness across sampling seasons in 2017.**

| Treatments | Arborous layer | | | Shrub layer | | |
|---|---|---|---|---|---|---|
| | Species richness | Shannon-Wiener index | Pielou evenness index | Species richness | Shannon-Wiener index | Pielou evenness index |
| PF | $3.33 \pm 0.47^a$ | $0.33 \pm 0.13^a$ | $0.27 \pm 0.09^a$ | $12.00 \pm 0.82^a$ | $2.34 \pm 0.05^a$ | $0.87 \pm 0.03^a$ |
| MF | $6.67 \pm 2.49^b$ | $0.93 \pm 0.34^a$ | $0.49 \pm 0.1^a$ | $11.67 \pm 2.49^a$ | $2.30 \pm 0.23^a$ | $0.94 \pm 0.01^a$ |

| Herbaceous layer | | | Litter thickness across sampling seasons (cm) | | | |
|---|---|---|---|---|---|---|
| Species Richness | Shannon-Wiener index | Pielou evenness index | April | June | August | October |
| $22.33 \pm 2.87^a$ | $1.91 \pm 0.10^a$ | $0.89 \pm 0.03^a$ | $5.59 \pm 0.33^a$ | $4.7 \pm 0.24^a$ | $6.5 \pm 0.14^a$ | $7.67 \pm 0.25^a$ |
| $7.67 \pm 1.25^b$ | $0.76 \pm 0.20^a$ | $0.55 \pm 0.20^a$ | $7.07 \pm 0.11^b$ | $6.29 \pm 0.22^b$ | $7.75 \pm 0.2^b$ | $11.12 \pm 0.7^b$ |

Note:
Different lowercase letters indicate significant differences between the two forest types ($p < 0.05$).

## Plant species diversity indexes

Biodiversity indexes and litter thickness of arbor, shrubbery and grass vegetation communities are shown in Table 2. Three general diversity indexes are selected for calculation and analysis of plant diversity (*Zhou et al., 2021*; *Li et al., 2019*): Species Richness ($S$) (1), Shannon-Wiener Index ($H'$) and Pielou Evenness Index ($J$) (3). The calculation formulas are:

$$S = \text{plant species in the sample plot} \tag{1}$$

$$H' = -\sum_{i=1}^{S} P_i \ln P_i \tag{2}$$

$$J = H'/\ln S \tag{3}$$

where, $P_i$ is the ratio of the importance value of the i-th species to the total importance value of all species in the sample plot, importance value of shrub layer = (relative significance + relative density + relative frequency)/3, and importance value of herb layer = (relative height + relative density + relative frequency)/3.

## Soil sampling and physicochemical analyses

Soil samples (0–50 cm depth) were collected with an auger (10 cm) on April 20, June 20, August 20 and October 20 of 2017 (*i.e.*, the second year since the beginning of the experiment). Nine soil cores for each 10 cm (0–50 cm, 10 cm per layer) soil samples were randomly taken from each plot, and all nine samples from the same depth were mixed into one composite sample, and 30 (2 treatments × 3 repeats × 5 soil layers) soil samples were collected in each season. Samples were stored at 4 °C in plastic bags for a few days after collection. To homogenize the soil material, the humus samples were sieved through a 2-mm sieve. This method also removes live roots, mycorrhizal mycelia and coarse plant remnants. Then, within 72 h, the soil samples were taken to the laboratory and

divided into three parts. One part, for chemical analysis, was air dried through a 0.149 mm sieve and stored at room temperature before chemical analyses for SOC and STN, and then a 2 mm sieve for pH analysis. Another part was stored at 4 °C until determination of DOC and MBC content determination and 60-day cumulative C mineralization. The remaining part was frozen at −80 °C.

Soil moisture was determined after being oven-dried at 105 °C for over 24 h. Air-dried soil samples that had been passed through a 2 mm sieve were used for soil pH determination, using a pH meter (Sartorius PB-10) and a 1:2.5 soil-water mixture. In each plot, a HOBO UTBI-001 waterproof temperature data logger (Onset Computer Corp., Bourne, MA, USA) was embedded 10 cm underground in the soil. Plant litter was removed before the UTBI-001 was placed, and then the logger was covered with the same litter. Temperature was logged automatically every hour from April 20, 2017 to October 20, 2017 with over 9,000 soil temperatures collected for each plot.

## Soil C and N analyses

Total SOC and total N concentrations in the samples were measured directly by an elemental analyzer (Thermo Scientific FLASH 2000 CHNS/O; Thermo Fisher Scientific, Waltham, MA, USA) from a subset of air-dried samples which were passed through a 0.149 mm soil sieve. Data for active C components were collected as previously described in *Ma et al. (2022)*. Specifically, MBC concentration was measured using a $CHCl_3$–fumigation extraction technique: $10 \pm 0.5$ g of fresh soil was fumigated with $CHCl_3$, extracted with 40 mL of 0.5 mol $L^{-1}$ $K_2SO_4$, shaken for 1 h at 350 r $min^{-1}$, and then filtered through a 0.45 μm membrane after centrifugation for 5 min at 3,000 r $min^{-1}$. The concentration of the filtrate was quantified using a total organic C analyzer (Multi N/C 3000; Analytik Jena, Jena, Germany). DOC concentration was measured as the C concentration of non-fumigated soil samples (*Boyer & Groffman, 1996*). MBC was calculated as MBC = $E_C/k_{Ec}$, where $E_C$ represents the difference between fumigated and unfumigated soils extractable organic C and $k_{Ec} = 0.45$.

Total stocks of N and C, as well as active C and N component stocks, were calculated using the formula:

$$\text{Stock (kg m}^{-2}) == \sum_{i=1}^{n} \text{Concentration} \times \text{BD}_i \times h_i \times 0.01 \tag{4}$$

Where concentration (g $kg^{-1}$) is the total stock of N and C, as well as active C and N stock in layer i. $BD_i$ (g $cm^{-3}$) is the soil bulk density in layer. $h_i$ (cm) is the soil layer thickness, and n is the number of soil layers (*Schrumpf et al., 2011*).

## Soil C incubation

SOC mineralization was measured by the lye absorption method (*Zhen et al., 2019*). Fifty grams of fresh soil were incubated in a 300 mL sealed the container in a dark incubator at $25 \pm 1$ °C. The $CO_2$-C emitted from soils over the incubation period (7, 14, 21, 28, 35, 42, 49 and 56 days) was trapped in 0.1 mol $L^{-1}$ NaOH. The molarity of the resulting NaOH was determined by titration with 0.05 mol $L^{-1}$ HCl after carbonate was precipitated

with 1 mL of 1 mol $L^{-1}$ $BaCl_2$. The cumulative $CO_2$-C was calculated based on the cumulative production of $CO_2$ from the soils during the 56-day incubation period and was expressed as milligrams of $CO_2$-C per kilogram of dry soil. The 56-day cumulative C mineralization (mg C $kg^{-1}$; CMC) represents the cumulative amount of C mineralized at the end of the incubation. Mineralized carbon (MC) is expressed on a per kg soil C basis rather than per kg soil, which is intended to normalize C across soil type (total C) differences between the two forest types. Soil mineralized C at each time point is given by the equation:

$$\text{Mineralized C (mg kg}^{-1}) = C_{HCI} \times (V_0 - V_1) \times 22/0.03. \tag{5}$$

where C is the SOC content (g $kg^{-1}$). $C_{HCI}$ is the concentration of HCI (mol $L^{-1}$). $V_0$ is the volume of the blank titration (mL), and $V_1$ is the volume of HCI consumed (mL).

## Statistical analysis

SPSS 20.0 (IBM, Chicago, IL, USA) was used for statistical analyses. Each plot was considered as an experimental unit, and the replicated data (5 soil layers * 3 replicated plots) were averaged by plots for each analysis. Prior to conducting ANOVA, all variables were checked for normal distributions (Kolmogorov-Smirnov test) and homogeneity (Levene's test). Then, to test the effects of forest type, sampling season, soil depth, and their interactions on SOC, STN, DOC, DON, MBC, MBN, SM, and soil pH, we ran a three-way analysis of variance (ANOVA) for each response variable. Significant models were then examined with a *post hoc* Tukey's test. To explore the effects of forest type on C components and environmental factors in certain season, comparisons of the soil factors, including C and N components, pH, and SM, in the same season, were compared *via* by one-way ANOVA. All results are represented as mean values ± standard error, with the statistical significance calculated at the $p < 0.05$ level. We tested for the impact of forest type within a season with a student's t-test. To examine the relationships between soil chemical variables, the collected data were pooled from five soil depths among five sampling seasons and six independent plots ($n = 150$). Pearson relationships were examined using the "Performance Analytics" package in R (*R Core Team, 2020*) for visualization.

## RESULTS

### Bioenvironmental factors

Generally, the sites were similar in elevation, slope, aspect and bulk density (Table 1). Tree age and height (PF: 15.99 ± 0.81 m; MF: 15.18 ± 0.63 m) were similar in both forest types, but the difference in DBH was significantly greater (26%) in PF (19.48 ± 0.20 cm) than MF (15.12 ± 1.17 cm) (Table 1).

As shown in Table 2, in the arborous layer, plant diversity indexes of MF, *i.e.*, species richness (PF: 3.33 ± 0.47; MF: 6.67 ± 2.49), Shannon-Wiener index (PF: 0.33 ± 0.13; MF: 0.93 ± 0.34) and Pielou evenness index (PF: 0.27 ± 0.09; MF: 0.49 ± 0.10), were greater than that of PF. In the herbaceous layer, plant diversity indexes of MF, *i.e.*, species richness

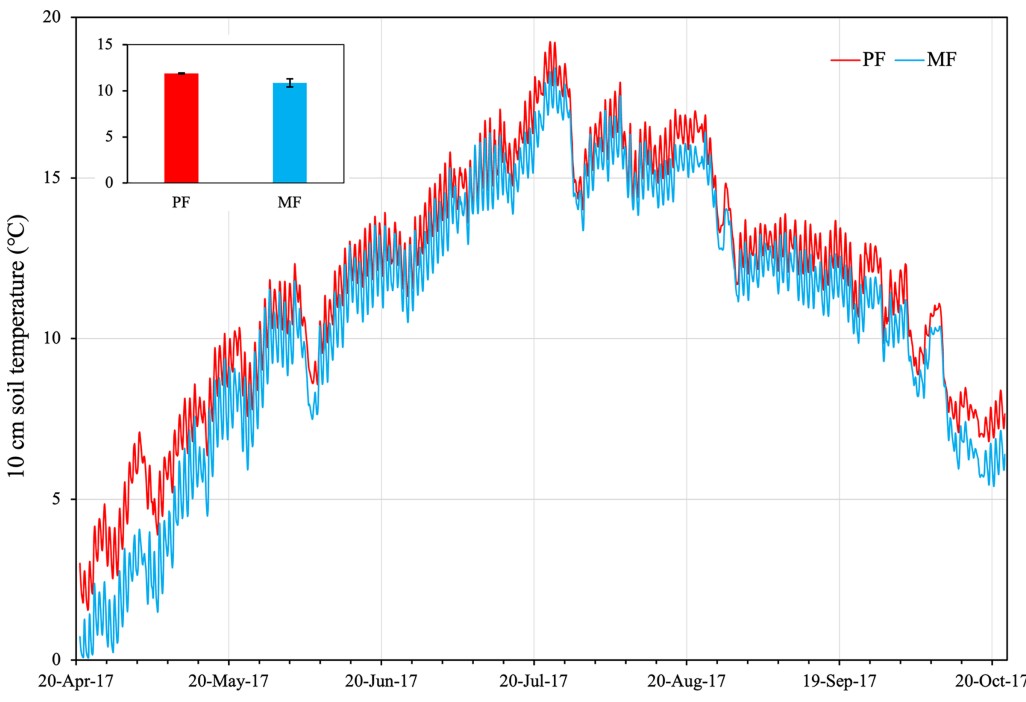

**Figure 1 Soil temperatures at 10 cm depth in plots of pure forest (PF) and mixed forest (MF) across the growing season in 2017.**

(PF: 22.33 ± 2.87; MF: 7.67 ± 1.25), Shannon-Wiener index (PF: 1.91 ± 0.10; MF: 0.76 ± 0.20) and Pielou evenness index (PF: 0.89 ± 0.03; MF: 0.55 ± 0.20), were smaller than that of PF. Both species richness significantly differed ($p < 0.001$) between the PF and MF. In addition, three plant diversity indexes of the shrub layer did not differ significantly (Table 2).

Litter thickness in MF was 31.8% thicker than that of PF throughout the 2017 growing season. Further, the litter thickness of MF was significantly thicker than that of PF in four sampling seasons, especially in October 2017, when litter was 45.1% thicker in MF plots (11.12 ± 0.7 cm) than PF (7.67 ± 0.25 cm) (Table 2).

## Physicochemical environmental factors

Soil temperature (ST) across the growing season averaged 11.88 ± 0.05 °C (STD from three plot repeats) in PF plots, which was significantly higher ($p < 0.05$) than in MF (averaged 10.87 ± 0.36 °C) (Fig. 1). From April to August, ST of the two forest types generally rose, reaching a peak value on July 24 (PF: 19.21 °C, MF: 18.3 °C). From May 20 to October 03, ST did not differ significantly between the two types (PF: 13.81 ± 0.16 °C; MF: 13.04 ± 0.16 °C) (Fig. 1). From October 03 to 22, ST differed significantly in the two types with PF (8.68 ± 0.29 °C) > MF (7.79 ± 0.11 °C).

Across the five sampling seasons, soil pH value in PF (7.00 ± 0.18) was 9% higher than in MF (6.4 ± 0.24) (Fig. 2A). ANOVA indicated that pH differed significantly between the two forest types ($p < 0.001$), among the seasons ($p < 0.001$) and at various soil depths ($p < 0.001$) (Table 3). When analyzed separately in each sampling season, pH was

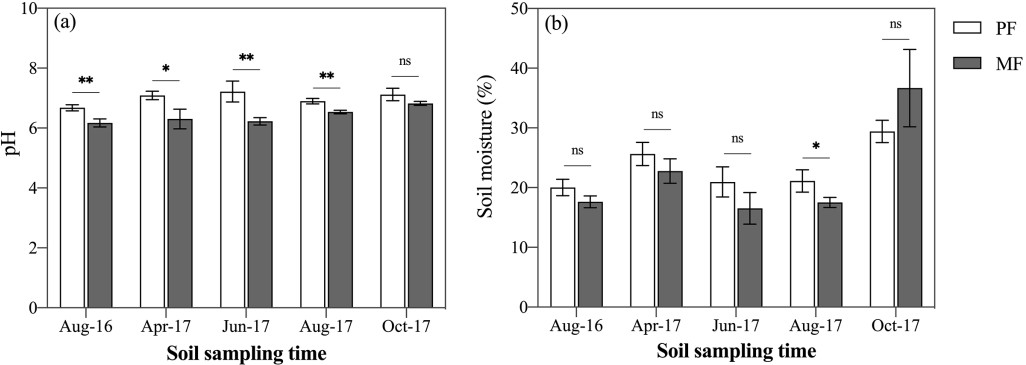

**Figure 2 Variation in soil pH (A) and soil moisture (B) in pure forest (PF) and mixed forests (MF) across the growing season in 2017 and August 2016.** Each value in the figure represents the average value of three plot replicates from five soil depths. The error bars represent the standard error and * indicate significant differences among treatments. ns $p > 0.05$; *$p < 0.05$; **$p < 0.01$.

**Table 3 Results of the three-way ANOVA for the soil active carbon and the soil properties among forest type, sampling season and soil depth in 2017.**

| Factors | MS | | pH | | SOC stock | | STN stock | | SOC/STN | | DOC stock | |
|---|---|---|---|---|---|---|---|---|---|---|---|---|
| | F | p | F | p | F | p | F | p | F | p | F | p |
| Tr | 3.907 | 0.051 | 195.864 | <0.001 | 31.867 | <0.001 | 18.638 | <0.001 | 0.113 | 0.738 | 47.163 | <0.001 |
| Sea | 80.423 | <0.001 | 17.063 | <0.001 | 4.043 | 0.004 | 4.619 | 0.002 | 13.962 | <0.001 | 115.907 | <0.001 |
| Dep | 40.072 | <0.001 | 25.354 | <0.001 | 48.073 | <0.001 | 45.782 | <0.001 | 1.896 | 0.117 | 8.612 | <0.001 |
| Sea * Tr | 12.244 | <0.001 | 9.809 | <0.001 | 1.501 | 0.208 | 0.405 | 0.804 | 3.899 | 0.006 | 3.233 | 0.015 |
| Tr * Dep | 0.708 | 0.588 | 3.107 | 0.019 | 2.331 | 0.061 | 1.673 | 0.162 | 1.938 | 0.110 | 0.569 | 0.686 |
| Seas * Dep | 1.642 | 0.072 | 0.875 | 0.599 | 0.874 | 0.601 | 1.488 | 0.119 | 0.916 | 0.554 | 1.017 | 0.446 |
| Seas * Tr * Dep | 0.374 | 0.986 | 1.341 | 0.188 | 1.201 | 0.281 | 0.557 | 0.908 | 1.001 | 0.462 | 0.214 | 0.999 |

| DON stock | | DOC/DON | | MBC stock | | MBN stock | | MBC/MBN | | MC | | CMC | |
|---|---|---|---|---|---|---|---|---|---|---|---|---|
| F | p | F | p | F | p | F | P | F | p | F | p | F | p |
| 3.909 | 0.051 | 69.799 | <0.001 | 0.573 | 0.451 | 0.409 | 0.524 | 3.162 | 0.078 | 56.782 | <0.001 | 119.997 | <0.001 |
| 395.827 | <0.001 | 199.267 | <0.001 | 78.175 | <0.001 | 25.126 | <0.001 | 31.445 | <0.001 | 90.412 | <0.001 | 506.965 | <0.001 |
| 13.253 | <0.001 | 1.622 | 0.175 | 3.116 | 0.018 | 11.410 | <0.001 | 4.415 | 0.003 | 13.460 | <0.001 | 155.475 | <0.001 |
| 2.656 | 0.037 | 3.577 | 0.009 | 3.916 | 0.005 | 1.053 | 0.384 | 3.904 | 0.005 | 0.553 | 0.648 | 1.577 | 0.201 |
| 1.134 | 0.345 | 1.752 | 0.145 | 3.776 | 0.007 | 0.534 | 0.711 | 1.334 | 0.263 | 8.196 | <0.001 | 3.592 | 0.010 |
| 2.588 | 0.002 | 1.465 | 0.128 | 1.020 | 0.442 | 1.520 | 0.108 | 1.803 | 0.041 | 1.192 | 0.303 | 3.279 | 0.001 |
| 0.638 | 0.845 | 0.659 | 0.827 | 0.488 | 0.948 | 0.729 | 0.759 | 2.293 | 0.007 | 1.886 | 0.049 | 0.859 | 0.591 |

**Note:**
Tre, two forest treatments; Sea, five sampling seasons; Dep, five soil depths, 10 cm per soil layer.

significantly lower in MF during four out of the five sampling seasons ($p < 0.05$). October 2017 was the exception ($p = 0.078$, Fig. 2A).

Generally, the average soil moisture (SM) was 24.80 ± 3.30% and 23.31 ± 7.19% in PF and MF plots, respectively (Fig. 2B). The sampling seasons and soil depths significantly affected SM ($p < 0.001$), while forest type had no significant effect on SM ($p = 0.051$) (Table 3). When analyzed separately in each sampling season, SM of PF was higher than

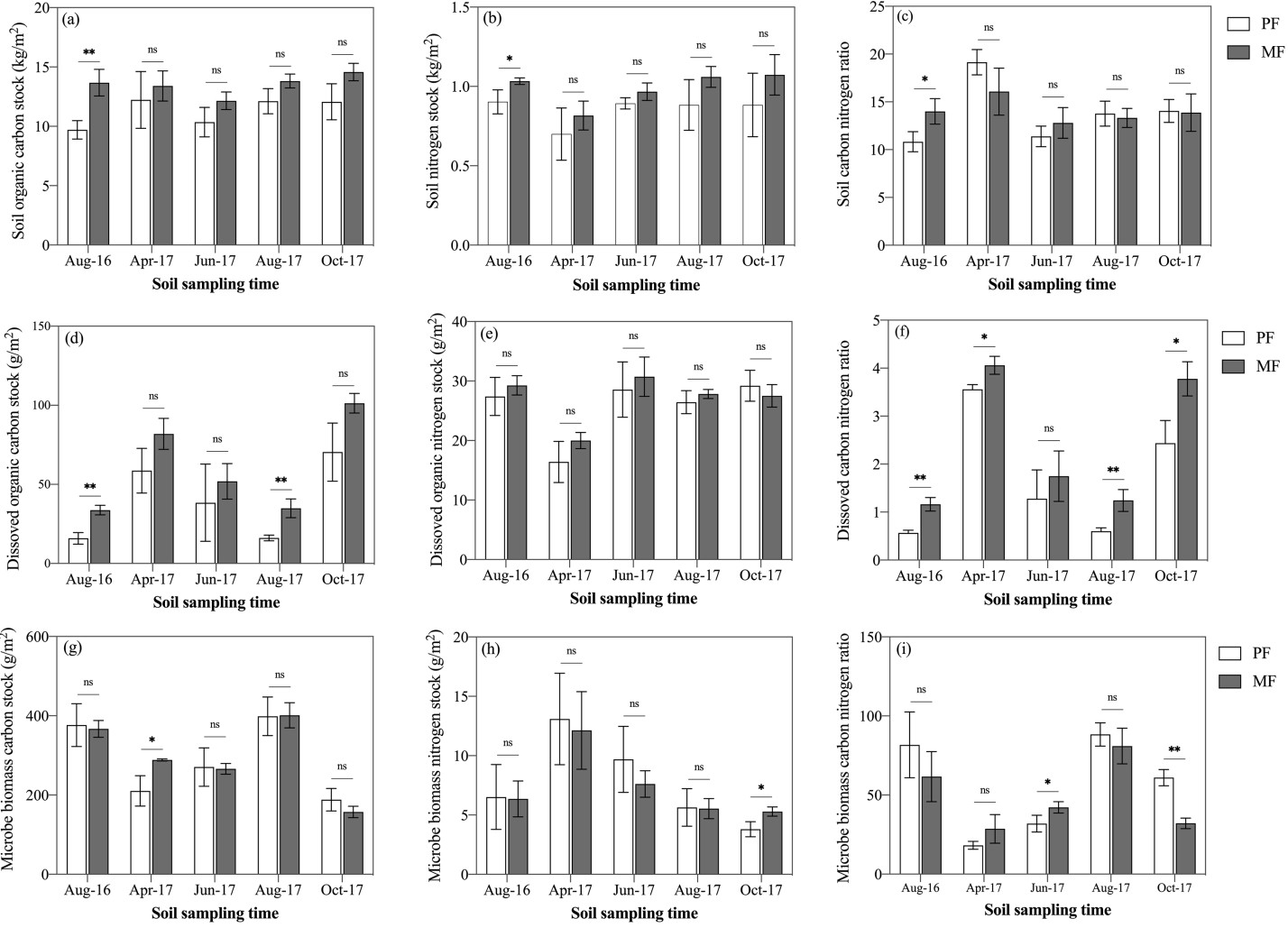

**Figure 3** Variation in SOC stock (A), STN stock (B), SOC/STN ratio (C), DOC stock (D), DON stock (E), DOC/DON ratio (F), MBC stock (G), MBN stock (H), and MBC/MBN ratio (I) in pure forests (PF) and mixed forests (MF) across the 2017 growing season and August 2016. SOC, soil organic carbon; STN, soil total nitrogen; DOC, dissolved organic carbon; DON, dissolved organic nitrogen; MBC, microbial biomass carbon; MBN, microbial biomass nitrogen. Each value in the figure represents the average value of three plot replicates. The error bars represent the standard error and * indicate significant differences among treatments, ns $p > 0.05$; *$p < 0.05$; **$p < 0.01$.     

that of MF except October 2017, with a significant difference ($p = 0.038$) only in August 2017 (Fig. 2b). Soil bulk density and mechanical composition were not found to differ significantly between the two forest types (Table 1).

## Soil organic carbon and nitrogen stocks

SOC stock ranged from 9.70 ± 0.64 kg C m$^{-2}$ to 14.57 ± 0.60 kg C m$^{-2}$ during the five sampling seasons for PF and MF (Fig. 3A). Average SOC stock of MF was 20% higher than that of PF (MF: 11.29 ± 1.18 kg C m$^{-2}$; PF: 13.52 ± 0.88 kg C m$^{-2}$) (Fig. 3A). Three-way ANOVA analyses indicated that the forest type ($p < 0.001$), sampling season ($p < 0.01$) and soil depth ($p < 0.001$) affected SOC stock stronger than their interactions (Table 3). When analyzed separately within each sampling season, SOC stock in MF

was higher than that in PF, but not significantly throughout the five sampling seasons, except for August 2016 (Fig. 3A). The tendency of soil total nitrogen (STN) stock was similar to SOC stock. Average STN stock was 16.5% higher in MF than PF (MF: 0.85 ± 0.09 kg C m$^{-2}$; PF: 0.99 ± 0.11 kg C m$^{-2}$) (Fig. 3B).

## Soil dissolved organic C and N stocks

DOC stock was affected by all three factors: forest type, sampling season and soil depth ($n$ = 150), though no interaction effects were found to affect DOC ($p$ > 0.05) (Table 3). DOC stock was significantly affected by the sampling season ($p$ < 0.001), *i.e.*, autumn (MF: 101.30 ± 6.17 g m$^{-2}$; PF: 70.41 ± 18.40 g m$^{-2}$) > spring (MF: 81.90 ± 9.79 g m$^{-2}$; PF: 58.65 ± 14.13 g m$^{-2}$) > summer (MF: 43.44 ± 6.21 g m$^{-2}$; PF: 27.27 ± 12.21 g m$^{-2}$) (Fig. 3D). Averaged over the various sampling seasons at 0–50 cm soil depths, DOC stock of MF was 52.3% higher than that of PF (MF: 60.77 ± 29.86 g m$^{-2}$; PF: 39.90 ± 24.62 g m$^{-2}$) (Fig. 3D). DOC stock in MF was higher than that of PF, but the difference was significant only in August 2016 and August 2017 (Fig. 3D). DON stock was not affected by forest type, but was affected by sampling season and soil depth ($p$ < 0.001) (Table 3). DOC/DON were affected by forest type and seasonal variation.

## Soil microbe biomass C and N stocks

MBC and MBN stock were affected significantly by sampling season ($p$ < 0.001) and soil depth ($p$ < 0.05), but not by forest type ($p$ = 0.451 for MBC, $p$ = 0.524 for MBN). A significant interaction effect was found for MBC stock between treatments and both season and depth ($p$ < 0.01; Table 3). When the data were taken from all sampling seasons, MBC stock of MF was only 2% higher than that of PF (PF: 288.80 ± 85.3 g m$^{-2}$; MF: 295.99 ± 85.2 g m$^{-2}$). In contrast, with the pattern of DOC stock, MBC stock had a maximum in August 2017 (MF: 401.27 ± 31.69 g m$^{-2}$; PF: 398.74 ± 48.76 g m$^{-2}$) and a minimum in October 2017 (MF: 157.28 ± 14.31 g m$^{-2}$; PF: 188.16 ± 28.47 g m$^{-2}$) (Fig. 3G). MBN stock decreased gradually from the beginning (April) to the end (October) of the growing season (Fig. 3H). The ratio continued to increase until August in both treatments, though the ratio in MF was lower than PF during August and October (Fig. 3I).

## Soil mineralization C

After incubation, CMC was found to differ significantly between the two forest types (Figs. 4A–4D; Table 3; $p$ < 0.05). In the two forests, CMC changed an average of 4.67 g C kg$^{-1}$ soil in PF and 4.04 g C kg$^{-1}$ soil in MF over a two-month period (Fig. 4A), where CMC changed 14.5% more in PF than MF. Additionally, CMC differed by sampling season ($p$ < 0.001), similar to DOC stock, *i.e.*, April (PF: 6.30 ± 0.14 g kg$^{-1}$ soil, MF: 5.83 ± 0.15 g kg$^{-1}$ soil) > October (PF: 4.84 ± 0.04 g kg$^{-1}$ soil, MF: 4.26 ± 0.04 g kg$^{-1}$ soil) > June (PF: 4.10 ± 0.18 g kg$^{-1}$ soil, MF: 3.44 ± 0.05 g kg$^{-1}$ soil) > August (PF: 3.45 ± 0.11 g kg$^{-1}$ soil, MF: 2.64 ± 0.06 g kg$^{-1}$ soil).

On average MC differed between PF sites and MF sites across sampling seasons in 2017 significantly ($p$ < 0.001) (Table 3) with 29.73% more MC in PF (20.35%) than MF (15.69%). Similar to CMC and DOC, MC was more abundant in the beginning (April: PF:

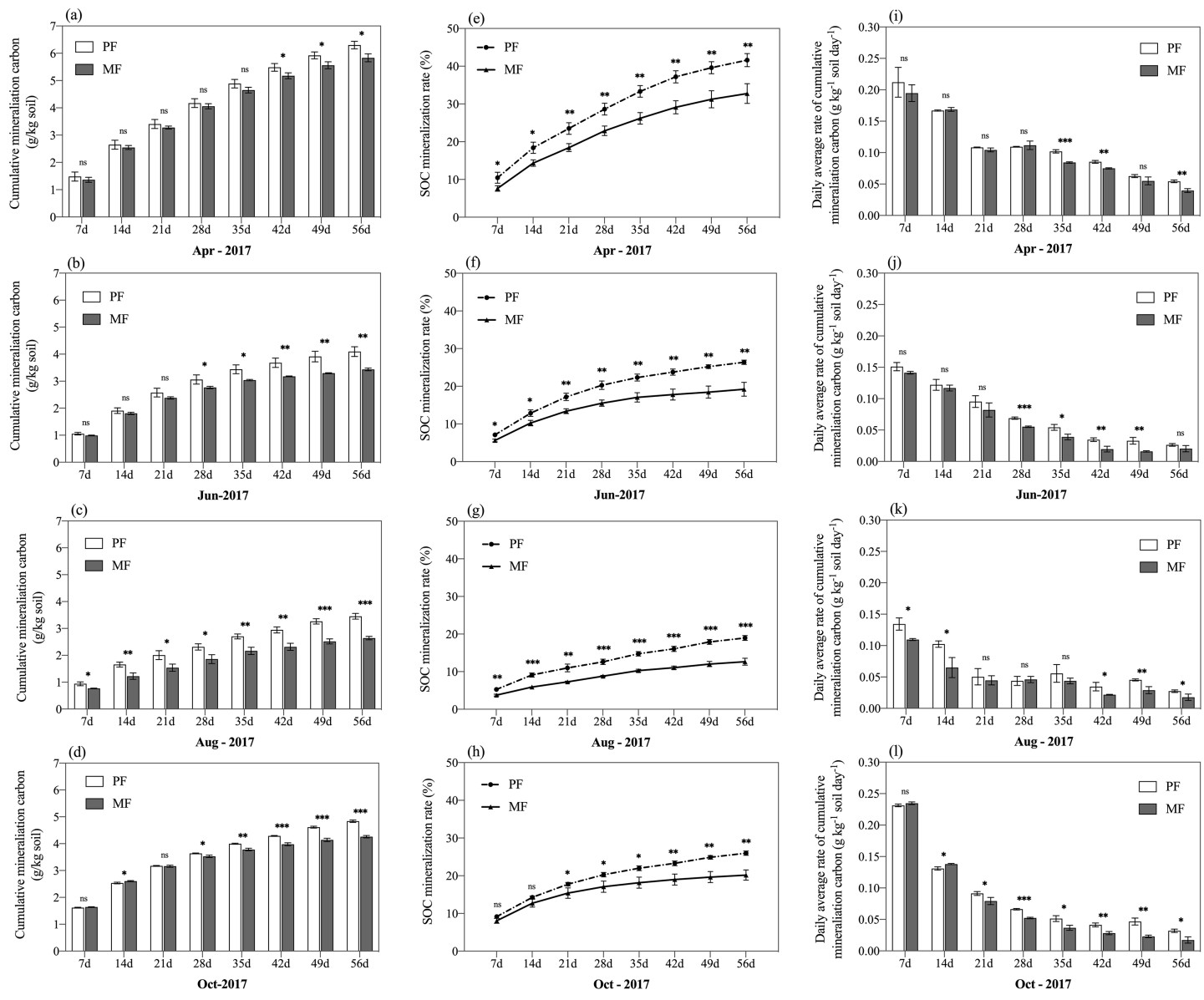

**Figure 4 Cumulative CO₂ emissions from soils (0–50 cm depth) over time (A) and SOC mineralization rate (B) from pure forests (PF) and mixed forests (MF) during the four sampling seasons every 56-day in April, June, August, and October 2017.** Systems with * differ significantly at ns $p > 0.05$; *$p < 0.05$; **$p < 0.01$; ***$p < 0.001$.

29.09%; MF: 22.83%) and end (October: PF: 19.70%; MF: 16.28%) of the growing season, reaching its lowest values in August (PF: 13.21%; MF: 8.97%) for both PF and MF (Figs. 4E–4H).

## DISCUSSION

Through monitoring the carbon (C) dynamics at five soil depths from 0–10 cm to 40–50 cm (five soil layers) and over five sampling seasons, SOC stock in coniferous-broadleaf mixed forests (MF) was found to be 20% higher than that of pure

forests (PF). To understand the mechanism driving the C dynamics, we considered soil depth, sampling season, soil physicochemical properties and active C components.

## Environmental factors drive C dynamics

Even though trees were taller in PF, we measured thicker litter throughout the growing season in MF and higher SOC stock in MF. Compared with coniferous forests of *Pinus tabulaeformis*, litters and fine roots of broad-leaved trees have lower C/N ratios, higher initial N content and faster microbial utilization, which was conducive to the improvement of soil active organic C, which partially explains our results (*Silver & Miya, 2001*). The Species richness index of PF understory vegetation was greater than that of MF, which may demonstrate tree species have a greater impact on soil C stock in forest ecosystems than undergrowth types (*Chen et al., 2005*; *Kraenzel et al., 2003*).

The significant difference in soil temperature (ST) between PF and MF throughout all sampling time suggested that this important environment factor (*Ma et al., 2010*) may be directly affected by the forest type. One explanation for the higher ST of PF was that the soil under MF receives less radiation than PF. Radiation, the main energy source, was also obstructed by more leaf litter in MF plots. Of the physical environmental factors, ST was the initial variable affecting both plant growth and soil C dynamics, and thus, C stock (*Todd-Brown et al., 2014*; *Falloon et al., 2011*). Both ST and MBC showed a single peak curve, because temperature can directly affect root respiration, microbial activity and decomposition of organic matter.

When forests were converted from pure forests into coniferous-broadleaf mixed forests, variation of soil moisture (SM) were influenced by many factors, including soil properties (*Gwak & Kim, 2017*), vegetation type (*Deng et al., 2016b*) and seasons (*Kumagai et al., 2009*). In this research, SM was significantly different with sampling season, soil depth and the interaction between season and forest type (Table 3). Rainfall can be intercepted by leaves, taken up by roots and lost in substantial amounts *via* evapotranspiration (*Jiménez et al., 2017*; *Jian et al., 2015*; *Yang et al., 2014*). Thicker litter partly explains the significantly lower moisture in MF, as SM in MF was measured to be significantly less than PF at the 30-50 cm soil layer and less water reached the deeper soil layers in the MF plots. Simultaneously, SM was strongly and positively correlated with DOC (Fig. 5), and DOC peaked in the later plant growing season, while SOC stock was also higher in MF than during the other three sampling seasons. Thicker litter accumulates in autumn and stimulates DOC leaching into the soil, making C available to microorganisms, which can effectively stimulate microorganisms involved in the mineralization of organic C and increase the mineralization rate, and further affect soil C stock (*Feng, Sun & Zhang, 2021*; *Deng et al., 2016a*).

Soil pH was one of the primary regulators of soil organic matter cycling (*Cheng et al., 2013*). pH was lower in MF across sampling season, and here soil pH negatively correlated with C and N ($p < 0.001$, $n = 150$, Fig. 5), suggesting than higher pH may decrease the capacity for SOC stock and nutrient supply (*Weil & Brady, 2016*). Moreover, we measured higher DOC stock in more acidic plots (lower soil pH) of MF, which may suggest high soil acidity increased SOC accumulation by inhibiting micro-bioactivity and accelerating the

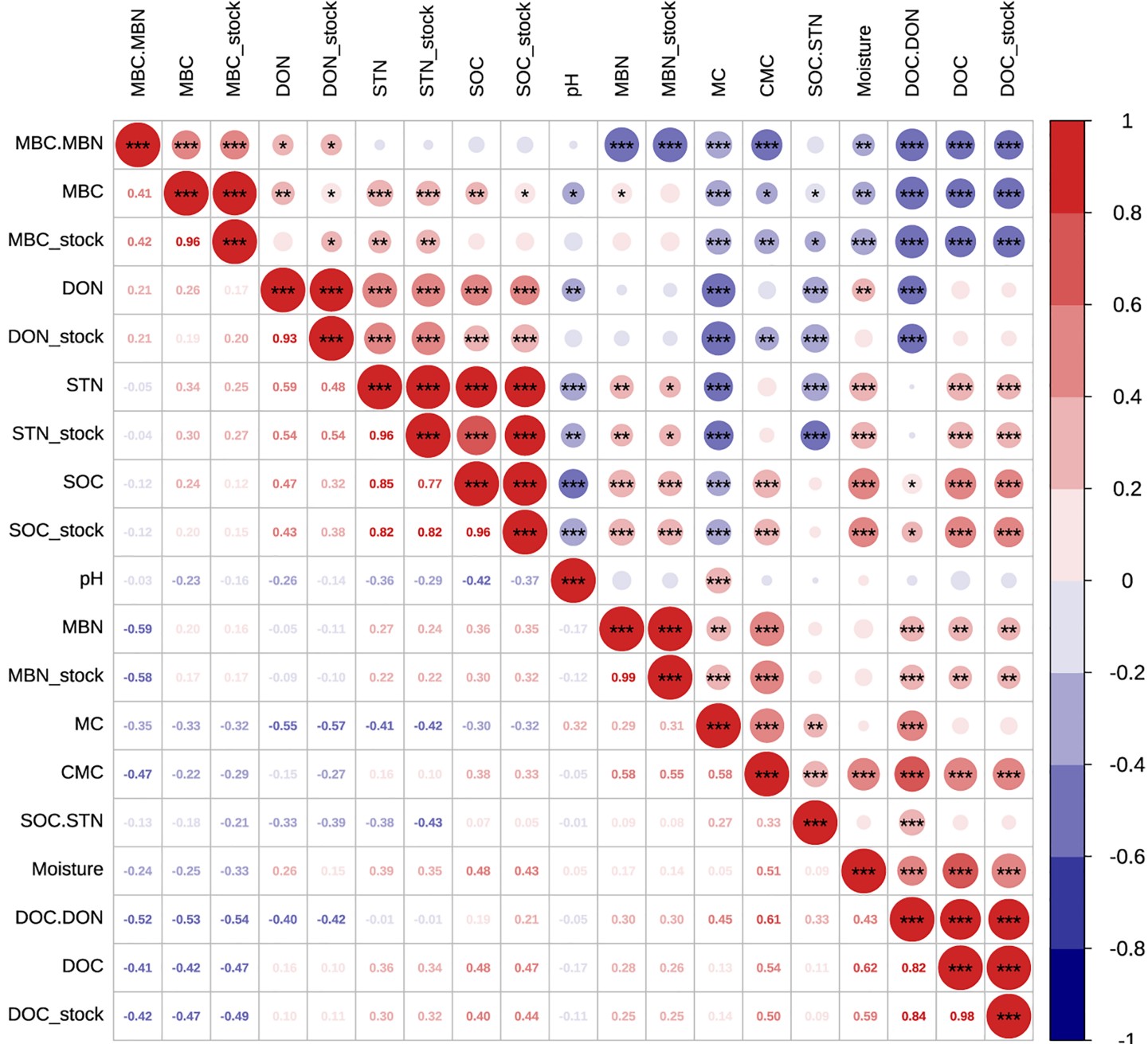

**Figure 5 Pearson relationships of different soil properties across both pure and mixed forests, sampling seasons and soil depths.** $n = 150$, *i.e.*, two forest type treatments × three repeats × five soil layers × five seasons. $^*p < 0.05$; $^{**}p < 0.01$; $^{***}p < 0.001$.

leaching of DOC into subsoils (*Funakawa et al., 2014*). However, MBC which relates to micro-bioactivity, did not differ significantly between PF and MF plots, probably because of the lower acidity and thicker litter in MF alleviates the inhibitory effect on soil microbes and thus achieves homeostasis.

## Activated soil C affects SOC stock

The active C stock was altered by either forest type or the interaction effect between soil depth and sampling season (Table 3). Ecologists have previously recognized that the quantity and quality of litter input has a significant impact on soil DOC accumulation (*Miao et al., 2019*; *Kalbitz et al., 2000*), and MF in this study had more branch leaves, plant residues, etc., and had a more proper conditions of C to leach into DOC stock. There was a significant seasonal pattern of DOC stock, *i.e.*, autumn > spring > summer, and the average content of the summer was 21.5% of the autumn, which were consistent with the findings of DOC fluxes during the growing season in a cool-temperature, broad-leaved deciduous forest in central Japan (*Chen et al., 2017*). DOC was labile and rapidly used and has been proposed to serve as an indicator of C availability for soil microorganisms (*Boyer & Groffman, 1996*). In this article, DOC stock was correlated significantly negatively with MBC stock ($R^2 = -0.49$) and significantly positively with SOC stock ($R^2 = 0.44$) and CMC ($R^2 = 0.50$) ($p < 0.001$; Fig. 5). When the biomass of microorganisms was large and activity was high, the microorganism's decomposition activity consumed more DOC, and simultaneously, soil mineralization also released more $CO_2$, which led to more SOC loss (*Iqbal et al., 2008*; *Lou et al., 2004*).

Soil MBC and MBN can both reflect the biomass and activity of soil microorganisms. Litter provided abundant C and nitrogen, which increased root growth and root exudates and promoted microbial growth and reproduction (*Miao et al., 2019*; *Tan & Chang, 2007*; *Hertel & Leuschner, 2002*). In this study, litter thickness in MF was significantly higher than that in PF, but no significant difference was measured in MBC and MBN, which may be related to the differences in understory vegetation and soil physicochemical properties (*Chen et al., 2005*). Studies have shown that soil MBC stock was significantly positively correlated with soil STN stock, indicating that the soil available nitrogen pool may be an important driving factor regulating MBC growth (*Wardle, 1992*). MBC stock had obvious seasonal changes, and the changes were characterized by a unimodal curve—an upward trend from April to August, and a downward trend from August to October (Fig. 3G). Studies have shown that seasonal changes in MBC stock of forests mostly depend on ST and SM in the temperate forest ecosystem of Uttarakhand, India (*Rawat, Arunachalam & Arunachalam, 2021*) and at the Guangxi Daguishan Forest station (*Wang et al., 2020*). MBC stock had a significantly positive correlation with ST and negative correlation with SM (Fig. 5).

Under identical temperature and water conditions in the laboratory, SOC mineralization reflects soil C availability and the differences of soil environmental factors in different forest types. Studies have shown that the main limiting factor affecting mineralization of SOC was the amount of soil active C and N (*Weintraub & Schimel, 2003*; *Yakovchenko, Sikora & Millner, 1998*). SOC stock was significantly and positively correlated with CMC ($R^2 = 0.33$, $p < 0.001$, $n = 120$), but was significantly and negatively correlated with MC ($R^2 = -0.32$, $p < 0.001$, $n = 120$) (Fig. 5), which showed that the more abundant the nutrient supply, the higher the microbial activity and the greater the potential mineralization capacity of the soil. Soil organic matter content was high, and the

substrates available to microorganisms increased, so the CMC was also high (*Davidson, Janssens & Luo, 2010*). Higher temperatures can promote SOC mineralization (*Laudicina et al., 2015*; *Xu, Inubushi & Sakamoto, 2006*), and increasing temperatures directly enhanced soil microbial activity and microbial respiratory entropy (*Verburg, Loon & Lükewille, 1999*), thus increasing the amount of $CO_2$ released (*Mayor et al., 2017*; *Stark et al., 2015*; *Fierer et al., 2005*). Higher temperatures in PF at 10 cm soil may cause higher MC, which increased C losses by $CO_2$ release. CMC was significantly positively correlated with SM ($R^2 = 0.51$, $p < 0.001$), DOC stock ($R^2 = 0.50$, $p < 0.001$), and significantly negatively correlated with MBC stock ($R^2 = -0.29$, $p < 0.01$). We also found that CMC was more abundant in April and October 2017, similar to when DOC stock was at its highest, but opposite to the seasonal trend of MBC stock. DOC consumption may be related to an increase in CMC. MC was significantly positively correlated with pH ($R^2 = 0.32$, $p < 0.001$), and soil organic matter was oxidatively decomposed by microorganisms or inorganic chemicals to generate $CO_2$ and carbonate. The latter may increase soil pH, which can partly explain the higher pH of PF. Mineralization is a biochemical process affected by the microbiological environment, temperature, soil moisture and active C or N components.

## CONCLUSION

In this study, we combined sampling plots and laboratory incubations to reveal SOC stock changes and the key factors affecting SOC stock in response to different forest types (PF: pure coniferous plantations *vs* MF: coniferous-broadleaf mixed plantations) on Taiyue Mountain, North China. We conclude that MF is able to store soil C better than PF, while the environmental factors and the active C stock, especially CMC, drives SOC stock dynamics. Environmental factors partly explained the C dynamics in which thicker litter may have contributed to more nutrient input, and lower soil temperature and pH may inhibit microbial decomposition in MF. Other active C components, like DOC or MBC, have opposite seasonal variation, with strong microbial activity and may make use of more leached DOC while releasing more $CO_2$. MBC was not enhanced in MF, and we suggest microbial communities may be a potential driver of the differences in MC between the two forest types, but that claim requires further study. Therefore, we recommend planning coniferous-broadleaf mixed plantations in a continental seasonal climate zone in North China to promote C retention and sequestration in response to climate change.

## ACKNOWLEDGEMENTS

We gratefully acknowledge the support from the Taiyue Forestry Bureau and the Haodifang Forestry Centre for fieldworks.

### Funding

This study was supported by the China Postdoctoral Science Foundation (Grant No. 2021M693035). The study was also supported by the National Key Research and Development Program of China (2016YFD0600205) and the National Key Research and

Development Program of China (2016YFC0501101-3). The funders had no role in study design, data collection and analysis, decision to publish, or preparation of the manuscript.

### Grant Disclosures
The following grant information was disclosed by the authors:
China Postdoctoral Science Foundation: 2021M693035.
National Key Research and Development Program of China: 2016YFD0600205.
National Key Research and Development Program of China: 2016YFC0501101-3.

### Competing Interests
The authors declare that they have no competing interests.

### Author Contributions
- Zhenzhen Hao conceived and designed the experiments, performed the experiments, analyzed the data, prepared figures and/or tables, authored or reviewed drafts of the article, and approved the final draft.
- Zhanjun Quan analyzed the data, authored or reviewed drafts of the article, and approved the final draft.
- Yu Han conceived and designed the experiments, analyzed the data, authored or reviewed drafts of the article, and approved the final draft.
- Chen Lv analyzed the data, prepared figures and/or tables, and approved the final draft.
- Xiang Zhao analyzed the data, prepared figures and/or tables, and approved the final draft.
- Wenjie Jing analyzed the data, authored or reviewed drafts of the article, and approved the final draft.
- Linghui Zhu performed the experiments, prepared figures and/or tables, and approved the final draft.
- Junyong Ma conceived and designed the experiments, performed the experiments, analyzed the data, prepared figures and/or tables, authored or reviewed drafts of the article, and approved the final draft.

### Data Availability
The raw data are available in the Supplemental File.

### Supplemental Information
Supplemental information for this article can be found online at http://dx.doi.org/10.7717/peerj.13542#supplemental-information.

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
