# Peer review of "Soil mineralized carbon drives more carbon stock in coniferous-broadleaf mixed plantations compared to pure plantations"

_PeerJ, doi:10.7717/peerj.13542_

## Round 0.1 · original submission · Major Revisions

Dear Dr Zhenzhen Hao,

We received two revisions to your paper. One recommended rejection, the second major revision. I also checked the manuscript and I also found weaknesses in your paper, particularly related to the experimental design which is not clear. Also, the part related to biodiversity is very weak, without indications about species found that you could conveniently place as supporting material.

I recommend a major revision of your paper, but please resubmit your paper only if you will be able to demonstrate that the experimental design was adequate. Otherwise, please consider carefully the reviewers’ suggestions and plan a more elaborated experiment in the future. Comparing the differences in soil C retention capacities is not trivial: I recommend taking advantage of past experiments and published results.

Sincerely,

Leonardo Montagnani

Reviewer 1 ·

Basic reporting

- Language needs improvement (In many cases it is almost impossible what authors are trying to say).
- Some scientific terminologies are incorrectly used.
- Figures and tables are not well formatted, some tables even went outside of the page!
- Not enough background of the topic to justify how the research fits into broader filed of knowledge.
- Authors completely failed to justify why this study is important both in terms of scientific advancement and practical application.

Experimental design

- Poorly designed experiment.
- Not enough replications (in fact no actual replication, only pseudo-replication).
- Statistics used on only ~3 data points barely carry any merit to justify a conclusion.
- Soil samples are not collected in a properly designed fashion (there are missing measurements in 2017). Besides, sampling schemes are not clearly described.
- Methods are not described with enough details to be replicated by someone not part of this experiment.

Validity of the findings

As I mentioned earlier, the authors failed to justify the novelty of this research. Moreover, the poorly designed experiment makes it almost impossible to draw any data-supported conclusion.

Additional comments

I have proved some comments in the attached document.

Annotated reviews are not available for download in order to protect the identity of reviewers who chose to remain anonymous.

Reviewer 2 ·

Basic reporting

I am not a native speaker.
The context is presented correctly.
The article contains the correct structure.
Table 2, Figures 2 and 3 – please add asterix according to the table footnote.

Experimental design

Please find suggestion in enclosed document.

Validity of the findings

Please find suggestion in enclosed document.

Additional comments

Please find suggestion in enclosed document.

Annotated reviews are not available for download in order to protect the identity of reviewers who chose to remain anonymous.

---

## Round 0.2 · Major Revisions

Dear Dr. Hao,

We received a full evaluation of your paper. I also checked your text and I fully support the indications done by the reviewer.

In addition, I warmly recommend considering the large uncertainty existing in soil C computations, which frequently prevents drawing conclusions from a simple comparison of different plots.

I, therefore, recommend considering the following paper before drawing any conclusions from your experiment: Schrumpf M, Schulze ED, Kaiser K, Schumacher, J. How accurately can soil organic carbon stocks and stock changes be quantified by soil inventories?. Biogeosciences. 2011; 8: 1193–1212. https://doi.org/10.5194/bg-8-1193-2011.

Sincerely,

Leonardo Montagnani

Reviewer 2 ·

Basic reporting

I am not a native speaker but the language and scientific terminology used still needs improvement.

Citation and references need checking – e.g. line 75 -Tian and Chang et al., while in the reference list there is Tan and Chang…; line 90 – lack of “and” (Sanford Kucharik, 2013).

Figures need improvement - details were shown the the report.

Experimental design

The sampling protocol is still not clearly explained.

Validity of the findings

Studies in the proposed scope are not new to the contribution of forest soils to C sequestration.

The data provided needs improvement, e.g., statistical significance. In many places in the text with results it is difficult to find the data in tables or graphs.

Additional comments

no comments

Annotated reviews are not available for download in order to protect the identity of reviewers who chose to remain anonymous.

---

## Round 0.3 · Major Revisions

Dear Dr. Hao and Dr. Ma,

We received one report for your article. While the reviewer finds some improvements, he/she also notes some relevant points to be improved.

Please answer to all the points raised by the reviewer and carefully consider these indications in the revised version of your manuscript.
Sincerely,

Leonardo Montagnani

Reviewer 2 ·

Basic reporting

I am not a native speaker but in my opinion, the article has been improved on the language side. Minor language changes were suggested. Citation – line 171 – Ma et al. instead of Ma ?; please check. Figures and Table have been improved. The article is structured correctly.

Experimental design

The sampling protocol has been improved, although there are aspects to be completed, e.g., lack of methodology to the data shown in Table 2.

Validity of the findings

Studies in the proposed scope are not new to the contribution of forest soils to C sequestration, however increase the number and area of forest soils examined. The data provided has been improved.

Additional comments

Overall the article has been improved in many ways. However, it still requires some revision before publication. In enclosed some suggestions for consideration.

Annotated reviews are not available for download in order to protect the identity of reviewers who chose to remain anonymous.

---

## Round 0.4 · accepted · Accept

Dear Dr. Hao and Dr. Ma,

I am pleased to inform you that I consider your paper acceptable now.

Thank you for considering PeerJ for publishing your study.

Sincerely,

Leonardo Montagnani